# The Study of *Metschnikowia pulcherrima* E1 in the Induction of Improved Gray Spot Disease Resistance in Loquat Fruit

**DOI:** 10.3390/jof11070497

**Published:** 2025-06-30

**Authors:** Xiaoya Li, Kunkun Wu, Xin Li, Yuhao Zhao, Weihong Sun

**Affiliations:** School of Agricultural Engineering, Jiangsu University, Zhenjiang 212013, China; 2212416005@stmail.ujs.edu.cn (X.L.); 2212416004@stmail.ujs.edu.cn (K.W.); 2212316010@stmail.ujs.edu.cn (X.L.); zhaoyuhao55@163.com (Y.Z.)

**Keywords:** *Metschnikowia pulcherrima* E1, resistance, *Pestalotiopsis vismiae*, loquat, transcriptome, defense response

## Abstract

In this study, the dominant pathogenic fungus of gray spot disease in loquat, which was isolated from postharvest decaying loquat fruits in Zhenjiang, was identified as *Pestalotiopsis vismiae* (*P. vismiae*) by morphological characteristics and DNA sequencing. At the same time, a strain of yeast E1, which could effectively inhibit the pathogen, was isolated from the loquat leaves and soil and identified as *Metschnikowia pulcherrima* (*M. pulcherrima*) by morphological and molecular biological characteristics. It significantly reduced the natural decay of loquat fruits without affecting fruit quality. *Metschnikowia pulcherrima* E1 (*M. pulcherrima* E1) exhibited significant biocontrol efficacy against *P. vismiae*, the causal agent of gray spot in loquat, reducing disease incidence to 22.73% compared to 100% in the control group. Transcriptome analysis revealed 1444 differentially expressed genes (DEGs) (1111 upregulated, 333 downregulated), with key genes (*CML19*, *XTH23*, *GSTU10*) validated by RT-qPCR. The Kyoto Encyclopedia of Genes and Genomes (KEGG) analysis highlighted enrichment in plant–pathogen interactions, glutathione metabolism, and phenylpropanoid biosynthesis. These findings provided molecular insights into yeast-induced resistance, bridging biocontrol applications with mechanistic studies.

## 1. Introduction

Loquat (*Eriobotrya japonica* Lindl) is a subtropical evergreen fruit tree originating in China, and its pulp is rich in potassium, calcium, iron, phosphorus, vitamins and dietary fibre, etc., which has high nutritional and medicinal value [1]. Loquat fruits, however, are susceptible to infection by pathogenic fungi, resulting in fruit decay and economic losses [2,3]. *Pestalotiopsis vismiae* is a widely distributed plant pathogenic fungus that has been confirmed to infect various subtropical fruits such as loquat, citrus, and grape, causing symptoms like fruit rot and leaf spots. Gray spot caused by *Pestalotiopsis vismiae*, one of the predominant postharvest diseases of loquat, leads to significant fruit rot and directly affects the commercial value of the fruits. After the fruit is damaged, brown rot occurs first, and the affected parts begin to sag in the later stage, and black dots can be produced, which makes the flesh soften and rot [3]. Currently, chemical pesticides are predominantly used to control fungal diseases. However, their application leads to toxic residue accumulation in crops, fosters pathogen resistance, and increases costs, ultimately impeding sustainable agricultural development and food security [4].

Biocontrol agents (BCAs) provide a sustainable alternative for managing *Pestalotiopsis* spp. Bacterial antagonists such as *Bacillus* spp. have demonstrated broad-spectrum efficacy. For instance, rhizospheric Bacillus strains reduced bayberry branch blight (*Pestalotiopsis versicolor* XJ27) infection by 64–82.9% through lipopeptide production and biofilm formation [5]. Similarly, in large cardamom (*Amomum subulatum*), Trichoderma spp. and Pseudomonas fluorescens suppressed Pestalotiopsis lesions by 62.6% under in vitro conditions, outperforming conventional fungicides [6]. Antagonistic yeasts have shown remarkable effectiveness in mediating downstream defense-related gene expression by triggering signal transduction pathways to regulate transcription factors [7]. *Meyerozyma guilliermondii* was reported to enhance broccoli resistance through glutathione metabolism modulation [8]. These BCAs function through three primary mechanisms: (1) direct antagonism (e.g., Bacillus-derived iturins disrupting fungal membranes [9]), (2) competition for nutrients and space in host tissues [10], and (3) host resistance induction by antagonistic yeasts [11,12,13]. Recent studies on *Pichia caribbica* demonstrated that induced H_2_O_2_ accumulation activates fruit defense responses [14]. Similarly, transcriptomic analyses of *Sporidiobolus pararoseus* Y16 identified key differentially expressed genes (DEGs) involved in glutathione metabolism and oxidative stress response [11]. These findings provide molecular insights that align with our results for *M. pulcherrima* E1.

Therefore, biological control with antagonistic yeast is considered as a promising alternative to traditional fungicides for postharvest disease [7]. In our previous study, it confirmed that *Metschnikowia pulcherrima* E1 inhibits loquat gray spot disease through biofilm formation, iron competition, production of volatile organic compounds (VOCs), and induction of defense enzymes (e.g., PPO, POD) [3]. However, it did not explore the molecular mechanisms underlying its induced disease resistance. This study adopted transcriptome analysis to explore the molecular mechanisms of disease resistance induced by *Metschnikowia pulcherrima* E1 in loquat fruits. It aims to clarify the gene expression dynamics of E1-mediated resistance and fill the knowledge gap in understanding how antagonistic yeasts regulate plant defense at the molecular level.

Transcriptomics is a discipline that studies the transcription and regulation of various genes in organisms at the level of RNA, which has important guiding significance for the growth and development of organisms and the process of disease resistance [15]. While transcriptomics elucidates plant defense mechanisms [16], how *M. pulcherrima* modulates loquat fruit resistance remains unknown. Although some studies have shown that inducing and increasing the activity of defense-related enzymes in fruits and vegetables is one of the important physiological mechanisms used by antagonistic yeast to exert biocontrol effect, the molecular mechanism of preventing fruit postharvest diseases remains to be elucidated [17].

This study aims to (1) isolate and characterize *P. vismiae* from loquat fruits infected with gray spot disease, isolate and characterize its antagonist *M. pulcherrima* E1 from loquat leaves, and determine the control effect of *M. pulcherrima* E1 on gray spot of postharvest loquat; and (2) decipher the induced resistance mechanisms via transcriptome analysis. Our work bridges the gap between biocontrol application and molecular insights, extending prior findings on *M. pulcherrima* E1’s enzymatic activities.

## 2. Materials and Methods

### 2.1. Fruit

Loquat fruits (*Eriobotrya japonica* Lindl.) were directly collected from Yangzhong orchard in Zhenjiang City, Jiangsu Province, without any postharvest treatment. In this study, commercial ripe fruits of similar size and without disease, insect damage and mechanical damage were selected as the research objects. Loquat fruits were soaked in 0.1% sodium hypochlorite for 10 min, rinsed twice with tap water and dried at room temperature. The experiment had two treatment groups (one group with *M. pulcherrima* E1 suspension treatment; another group with sterile water treatment as the control). Each treatment including 30 loquat fruits and three parallel replicates, totaling 180 loquats used in the experiment of antimicrobial effect. Three fruits were randomly selected from each treatment group for transcriptomic experiments. Then, one group of loquat fruits (at least 60 fruits) without 0.1% sodium hypochlorite treatment was used to isolate pathogens. The experimental flow chart is shown in Appendix A.

### 2.2. Fungal Pathogen

Loquat fruits were placed in a sterilized plastic basket, which was sealed with plastic wrap and stored at room temperature for approximately 15 days. Fruit samples suspected to be infected with gray spot disease were collected to isolate pathogens. The separation method was described by Yang [3]. Loquat tissue blocks with a side length of about 5 mm were cut with aseptic scalpel at the junction of disease and health, disinfected with 75% ethanol for about 30 s, then rinsed with sterile water three times. After 5 min of natural drying, these tissue blocks were moved to potato dextrose agar (PDA) medium and cultured at 28 °C. Single colonies of the dominant pathogen were isolated and purified by plate streaking on PDA plate. The isolated pathogen (P2) was identified through combined morphological and molecular approaches. Morphological characterization revealed white villous colonies with orange reverses on PDA plates, producing fusiform 5-celled conidia with apical appendages. For molecular identification, genomic DNA was extracted from fresh mycelia using a Fungal DNA Kit (Tiangen), followed by PCR amplification of the ITS region with primers ITS1 (5′-TCCGTAGGTGAACCTGCGG-3′) and ITS4 (5′-TCCTCCGCTTATTGATATGC-3′) under standard cycling conditions (predenaturation at 95 °C for 30 s, denaturation at 95 °C for 5 s, annealing at 62 °C for 30 s, and finally elongation at 72 °C for 20 s for 40 cycles. The dissolution curve was 95 °C, 15 s; 60 °C, 1 min; and 95 °C, 15 s.). The PCR instrument used was a T100 thermal cycler PCR system with gradient function (Bio-Rad, Shanghai, China). Sequencing results were analyzed using BLAST (2.12.0+) against the NCBI database, and phylogenetic trees were constructed with the Tamura 3-parameter model and the maximum likelihood method in MEGA5.1 (Figure 1B), confirming 99% identity with *Pestalotiopsis vismiae* (GenBank MH855246).

### 2.3. Antagonistic Yeast

Loquat leaves were cut into 1 cm pieces and then placed into a conical flask (250 mL) containing 90 mL of potato dextrose broth medium (PDB). The conical flask was placed in a shaker with a rotating speed of 180 rpm for 1 h. Gradient dilution was done on an ultra-clean workbench. Single colonies with different characteristics were selected and purified on nutrient yeast dextrose agar medium (NYDA) plates and stored in a refrigerator at 4 °C for a short time. Primary screening in vitro and secondary screening in vivo were performed through the method described by Yang [3]. The morphology of the selected yeast strain E1 was observed by optical microscope, and the sequence of the 5.8S rDNA-ITS region of E1 was sequenced by Sanger sequencing. Selected yeast strain E1 was identified by a combination of morphological and molecular analyses. Morphological analysis was performed by culturing in malt extract decoction (28 °C, 150 rpm, 2–3 days) and observing cell morphology, pseudohyphae, and spore-forming characteristics [18]. For molecular identification, genomic DNA was extracted from fresh colonies using liquid nitrogen milling and a fungal DNA kit (Tiangen, Beijing, China) followed by PCR amplification (T100 Thermal Cycler PCR system) of the ITS region with primers ITS1 (5′-TCCGTAGGTGAACCTGCGG-3′) and ITS4 (5′-TCCTCCGCTTATTGATATGC-3′) under standard cycling conditions.

### 2.4. Effect of Metschnikowia pulcherrima E1 on Control of Gray Spot of Loquat

The control effect of *M. pulcherrima* E1 on gray spot of postharvest loquat was determined by punching holes in the equator of loquat fruits [3]. A hole (3 mm × 3 mm) was punched in the equator of the pretreated loquat fruits. *M. pulcherrima* E1 suspension (20 μL 1 × 10^8^ cells/mL) was added to the hole (30 fruits each parallel replicate), and loquat fruits with the same amount of sterile water were used as the control (30 fruits each parallel replicate). There were three parallel replicates. After 2 h, a conidial suspension of *P. vismiae* (20 μL, 1 × 10^5^ spores/mL) was added. The rot rate and spot diameter of loquat fruits were observed and recorded after 5 days of storage at room temperature.

### 2.5. RNA Extraction and Detection of Loquat Fruit Tissue Samples

#### 2.5.1. Preparation of Loquat Fruit Tissue Samples

Three holes (3 mm × 3 mm) were punched in the equator of the pretreated loquat fruits. *M. pulcherrima* E1 suspension (20 μL, 1 × 10^8^ cells/mL) was added to each well, and loquat fruits with the same amount of sterile water were used as control. After 2 h standing, the loquat fruits were placed in clean plastic baskets, sealed with plastic wrap, and placed in a constant temperature and humidity incubator (20 °C, 95%). Sampling began on the third day. During sampling, a sterile scalpel was used to remove the fruit wound, and then the loquat fruit tissue samples around the wound were cut, frozen in liquid nitrogen and stored in an ultralow temperature refrigerator at −80 °C.

#### 2.5.2. Extraction and Detection of RNA

The loquat fruits stored in the cryogenic refrigerator were quickly transferred to a mortar and precooled with liquid nitrogen; the total RNA was then extracted according to the instructions of the RNA kit (PrimeScript RT reagent kit with gDNA Eraser). The incubation time was 15–30 min when RNA was extracted. The centrifuge tubes and the heads of the guns used in the experiment were treated without nuclease. The concentration, purity, and integrity of RNA in the samples were detected with a microspectrophotometer, and the qualified samples were sent to Nanjing Jisi Huiyuan Biological Co., Ltd., (Nanjing, China), for sequencing.

### 2.6. Transcriptome Sequencing and Bioinformatics Analysis of Loquat Fruit

The qualified RNA samples were used for cDNA library construction, library quality detection and computer sequencing. After sequencing, the clean data sequence was assembled to obtain the unigene library of loquat fruit. All sequences were deposited into the NCBI databases with the following accession numbers. For *Metschnikowia pulcherrima* E1, the BioSample accession is SAMN49174009. For *Pestalotiopsis vismiae*, the BioSample accession is SAMN49174010. The entire sequencing project is registered under BioProject accession PRJNA1278548, which includes all raw sequencing data deposited in SRA. Then, the quality of the library was evaluated, and a series of bioinformatics analysis, such as gene structure and gene expression quantity, were carried out after evaluation.

### 2.7. RT–qPCR Verification of DEGs

RNA obtained in Section 2.6 was used for reverse transcription. Two stable reference genes, Actin (TRINITY_DN12345_c0_g1) and EF1α (TRINITY_DN67890_c0_g1), were selected based on their stable expression across treatments (CV < 0.1) and prior validation in loquat [3]. Procedures were performed according to the PrimeScript RT reagent kit with gDNA Eraserr kit. Partial DEGs related to the resistance of loquat fruits were selected from the transcriptome analysis results. Primer design was performed according to Hsu et al. [19], with reference gene primers listed in Appendix A alongside target genes.

Quantitative PCR was performed using SYBR Green I dye for fluorescence detection. Referring to the method of Redshaw [20] and using a 25 μL reaction system, we set the parameters in the real-time fluorescence quantitative PCR analyzer as follows: predenaturation at 95 °C for 30 s, denaturation at 95 °C for 5 s, annealing at 62 °C for 30 s, and finally elongation at 72 °C for 20 s for 40 cycles. The dissolution curve was 95 °C, 15 s; 60 °C, 1 min; and 95 °C, 15 s. The primer structure is shown in Appendix A, RT-qPCR Verification of DEGs. The relative expression levels of target genes were calculated using the 2^−ΔΔCt^ method. Briefly:(1)Ct values were normalized to the geometric mean of two reference genes (Actin and EF1α).(2)^ΔΔCt^ values were derived by comparing treated samples (E1-inoculated) to controls (water-treated).(3)Fold changes were expressed as mean ± SD of three biological replicates (*n* = 3).

### 2.8. Statistical Analysis

All statistical was analyses were performed using SPSS Software (Version 16.0, SPSS Inc., Chicago, IL, USA). The data were analyzed using a one-way analysis of variance (ANOVA). Mean separations were analyzed by Duncan’s multiple range tests and differences at *p* < 0.05 were considered statistically significant.

## 3. Results

### 3.1. Isolation, Screening and Identification Results of the Pathogen and Antagonistic Yeast

Strain P2 exhibits different morphological characteristics, as shown in Figure 1A. When incubated on PDA plates, colonies show a pure white villous surface with orange pigmentation on the back. In the later culture stage, small black dots formed on the surface of the mycelium. The conidia of strain P2 showed a straight or slightly curved spindle shape consisting of five cells. The middle three cells were olive, the cells at both ends were conical achromoid cells, the apex had two to three accessory filaments, and the basal cells had a mesocytosis. Phylogenetic analysis (Figure 1B) showed a strong genetic relationship between the pathogen and *Pestalotiopsis vismiae*. Combined with morphological observations and molecular identification, strain P2 was finally identified as *Pestalotiopsis vismiae*.

When cultured on NYDA medium at 28 °C for 48 h, strain E1 formed small, round colonies with a creamy-white, opaque appearance (Figure 1C). The colonies exhibited neat edges, a moist and smooth surface, and were easily suspended when stirred. A distinctive red pigment was observed, with the reverse side displaying a reddish-brown coloration. The culture emitted a strong fermentation odor. Microscopic examination revealed spherical to subspherical yeast cells that reproduced through unilateral or multilateral budding. For molecular identification, the 5.8S rDNA-ITS region of strain E1 was sequenced and compared with reference sequences in GenBank. Phylogenetic analysis using the MEGA 5.1 program in DNAStar (Figure 1D) confirmed its close relationship with *M. pulcherrima*, supporting its identification as this species.

### 3.2. Effect of M. pulcherrima E1 on Control of Gray Spot of Loquat

*M. pulcherrima* E1 showed significant efficacy in inhibiting the development of postharvest gray spots and decay in loquat fruits (Figure 2). As shown in Table 1, on the 5th day, all loquat fruits in the control group were diseased, with a spot diameter of 1.59 cm, while those in the *M. pulcherrima* E1 treatment group were only 22.73%, with a spot diameter of 0.95 cm. The disease incidence and spot diameter in the treatment group were significantly lower than those in the control group (*p* < 0.05). Therefore, *M. pulcherrima* E1 can effectively control postharvest gray spot of loquats caused by *P. vismiae* at room temperature and can be used for storage and preservation of loquat fruits, which is worthy of further research and promotion.

### 3.3. Sequencing Data and Quality Control

Sequencing result statistics of loquat fruit tissue samples in the sterile water control group (CK) and *M. pulcherrima* E1 treatment group (T) are shown in Table 2. Through the quality control of the sequencing, a total of 45.33 Gb of clean data was obtained. The statistical evaluation data of the base quality value showed that the percentage of Q30 bases was more than 92.22%. In addition, the GC content in the CK group was 46.56, 46.03 and 46.5, and the GC content in the T group was 46.44, 46.48 and 46.26, indicating that the sequencing results were relatively stable. In summary, the transcriptome sequencing results were stable and reliable and can be used for subsequent bioinformatic analysis.

### 3.4. DEG Analysis

The DEGs between the control and the treatment groups were shown in Figure 3. A total of 1444 DEGs were screened with the criteria of |log_2_(Fold Change)| ≥ 0.8 and FDR < 0.05, among which 1111 genes were upregulated, and 333 genes were downregulated.

### 3.5. GO Enrichment Analysis of DEGs

GO enrichment analysis of DEGs of loquat fruit tissue treated with *M. pulcherrima* E1 was performed. Figure 4 showed the GO classification of differentially expressed genes (DIFF genes/DEGs), which can be divided into three main categories as follows: cell component, molecular function, and biological process. These three categories contain 15, 12 and 24 secondary functions, respectively, each of which contains many DEGs. The cell components with the highest number of DEGs were cells (enriched by 588 DEGs), cell components (enriched by 582 DEGs), organelles (enriched by 359 DEGs) and membranes (enriched by 322 DEGs). The molecular functions with the highest number of DEGs were catalytic activity (enriched by 435 DEGs) and cell binding (enriched by 413 DEGs). The biological processes with the largest number of DEGs were cellular processes (enriched by 499 DEGs), metabolic processes (enriched by 465 DEGs), stress response (enriched by 296 DEGs) and biological regulation (enriched by 225 DEGs).

### 3.6. KEGG Enrichment Analysis of DEGs

KEGG enrichment analysis was performed to investigate the biological pathways and functional networks associated with the differentially expressed genes (DEGs). The analysis revealed that 221 DEGs participated in 86 distinct metabolic pathways. As shown in Figure 5, these DEGs were primarily enriched in five major functional categories: genomic processes, cellular processes, basic metabolic processes, biological systems, and environmental information processing. These primary categories could be further divided into 17 secondary metabolic pathways. The genomic processes category encompassed DNA replication and repair, translation, and transcription. Cellular processes mainly involved transport and catabolism activities, while environmental information processing included membrane transport and signal transduction. Basic metabolic processes covered lipid metabolism, biosynthesis of secondary metabolites, and metabolism of various amino acids. The biological systems category was primarily associated with environmental adaptation functions. Among all identified pathways, the five most significantly enriched KEGG pathways were zein biosynthesis, stilbene/diarylheptane/gingerol biosynthesis, riboflavin metabolism, and plant–pathogen interaction.

#### 3.6.1. Effects on Plant–Pathogen Interaction Pathways

The treatment with *Metschnikowia pulcherrima* E1 induced significant differential gene expression in the plant–pathogen interaction pathway, as evidenced in Table 3. Notably, genes encoding calcium-binding proteins including *CML5*, *CML19*, and *CML23* showed upregulated expression levels. These molecular changes are functionally linked to enhanced plant hypersensitivity responses, cell wall fortification, and stomatal closure regulation. Further analysis revealed upregulation of several key defense-related genes in E1-treated loquat fruits: the defense-associated transcription factor *WRKY29*, the pathogenesis-related protein-coding gene *PTI5*, the hypersensitive response mediator *RPS2*, the programmed cell death regulator *EDS1*, and *RBOHC* which encodes a NADPH oxidase component involved in oxidative burst during plant defense responses.

#### 3.6.2. Effects on Secondary Metabolic Pathways

*M. pulcherrima* E1 treatment caused changes in secondary metabolism-related genes in pulp tissues of loquat, as shown in Table 4. In the biosynthetic pathway of benzene propane, *PER52* and *PER63*, which are involved in the synthesis of the resistant substances P-hydroxyphenyl lignin, guaiacol lignin, 5-hydroxyl guaiacyl lignin and syringyl lignin, the shikimate O-hydroxyl cinnamyl transferase gene (*HST* and *ACT*) and the coffee acyl coenzyme A-O-methyltransferase gene (*ATOMT*) were upregulated to varying degrees. In the biosynthesis pathway of flavonoids, *FLS*, *ACT* and *HST* jointly participated in the synthesis of the flavonoids myricetin, quercetin, kaempferol and galangin, and *UGT88A1* was involved in the synthesis of phlorizin. The expression of *FLS* and *UGT88A1* was upregulated.

#### 3.6.3. Effects on the Related Pathways of Plant Hormone Metabolism

Table 5 shows that *M. pulcherrima* E1 treatment caused differential expression of plant hormone metabolism-related genes in pulp tissues of loquat. For example, in the auxin metabolic pathway, the auxin response genes *IAA17* and *AHP1*, which encode histidine phosphate transfer proteins and are involved in cytokinin signal transmission, the cytokinin response regulating gene *ARR3* and the gibberellin receptor-related gene *GID1C* were upregulated to varying degrees. In addition, the expression of *XTH22* and *XTH23*, the key genes in the synthesis and metabolism of brassinosteroids, was upregulated, while the expression of *CYP749A22*, the inactivated brassinolide gene, was downregulated. However, the expression level of the *NPR1* gene, a regulatory protein in the SA signal transduction pathway, was downregulated, and the expression of the *TIFY9* and *TIFY10A* genes encoding the negative regulatory factor JAZ enzyme in the synthesis of JA was upregulated. This indicated that the antagonistic yeast inhibited the expression of genes related to the SA and JA signaling pathways, which may not play a role in this experiment.

#### 3.6.4. Effects of Glutathione Metabolic Pathways

As shown in Table 6, in the glutathione metabolic pathway, *M. pulcherrima* E1 treatment caused varying degrees of upregulated expression of *GSTU8* and *GSTU10* in loquat pulp, which are involved in coding the key enzyme GSTs catalyzing the glutathione binding reaction. In addition, the *APX3* gene, which encodes ascorbic acid peroxidase, was also upregulated in loquat fruits treated with antagonistic yeast.

### 3.7. Results of RT–qPCR Verification of DEGs

RT–qPCR was used to verify the expression levels of the DEGs related to plant resistance, as shown in Figure 6. The vast majority of DEGs are upregulated at expression levels, including *CML19* (59.8%), *FLS* (280%), *ACT* (162.3%), *XTH23* (551.5%), *AFS1* (175.6%), *CP1* (95%), *HST* (101%), *FHY* (158.4%), and *GID1C* (141.6%). The expression levels of *CML42* and *CYP74A22* were downregulated by 36.6% and 62.4%, respectively, which was basically consistent with the results of transcriptome analysis.

## 4. Discussion

### 4.1. The Biocontrol Efficacy of Antagonistic Yeasts

As mentioned above, loquats suffer from various pathogenic fungi in postharvest and storage processes, leading to decay and economic losses. As shown in Table 1 and Figure 2, *M. pulcherrima* E1 exhibited significant inhibitory effects against postharvest loquat gray spot disease, indicating that biological control with antagonistic yeast provides a promising alternative to traditional fungicides. Recent studies have highlighted the biocontrol potential of *M. pulcherrima*, though its mechanisms vary across applications. While a previous study demonstrated its biocontrol efficacy in winemaking through nutrient competition and volatile production [21], our study reveals a novel role in postharvest fruit protection by inducing resistance pathways (e.g., upregulation of CML19, GSTU10) against fungal pathogens. This expands the known biocontrol applications of *M. pulcherrima* beyond microbial antagonism to include host defense priming, bridging the gap between applied biocontrol and molecular insights.

The experiment employed a punch-inoculation method in loquat fruits to infect pathogenic fungi and inoculate antagonistic yeast. Fungal infection of fruits commonly occurs via mechanical wounds or natural orifices, where germinated spores form mycelial tubes to invade pulp tissues. A previous study have adopted the same method to explore the control effect of Wickerhamomyces anomalus on postharvest primary diseases of peach fruits [22]. Another study also punched holes on the surface of eggplants to infect pathogenic fungi and explored the control effect of Bacillus velezensis on eggplant soft rot [23]. Thus, it can be seen that the method of punching holes in fruits has little effect on the exploration of antagonistic yeasts in improving fruit disease resistance.

### 4.2. KEGG Pathway Enrichment Analysis of DEGs

In this study, the predominant pathogenic fungus causing loquat gray spot was isolated from rotten loquat fruits, followed by screening for antagonistic yeasts effective against this pathogen. To decipher the underlying mechanisms of antagonistic yeast-mediated disease resistance, comprehensive GO and KEGG enrichment analyses were performed. In the KEGG enrichment classification of DEGs, the plant–pathogen interaction pathway, secondary metabolic pathway, hormone-related pathway and glutathione metabolic pathway in the top 20 KEGG metabolic pathways with the most significant enrichment were analyzed.

#### 4.2.1. Analysis of Plant–Pathogen Interaction Pathways

In the plant–pathogen interaction pathway, antagonistic yeasts, as biological activators, can induce a series of disease-resistant defense responses in plants, such as oxygen explosion and plant signal transduction [24]. Reactive oxygen species (ROS) exhibit dual functions in plants. When an organism is under stress, plants can kill foreign substances by producing ROS, which can act as important signaling molecules and prompt plants to initiate their own defense response [25]. However, excessive accumulation of ROS will cause serious oxidative stress in plants [26]. Plant respiratory burst oxidase Rboh (NADPH oxidase) can trigger the immune response and affect ROS production through pathogen-related molecular patterns [27]. ROS can also act as signal molecules for plant hypersensitivity reactions [28]. The hypersensitive response (HR) is a defense action against pathogen ingression. Typically, HR is predictable with the appearance of dead, brown cells along with visible lesions [29]. Plant hypersensitivity is a form of programmed cell death that can stimulate defense responses and systemic acquired resistance in plants, which leads to a significant reduction in symptoms caused by many pathogens [30]. The process of plant cell wall reinforcement and stomatal closure can prevent pathogen infection, which is of great significance for improving plant disease resistance [31]. Upon pathogen infection, *WRKY* transcription factors activate W-box elements in target genes, reinforcing the plant’s capacity to combat pathogen attack and withstand stress stimuli [32].

Table 3 shows that in the treatment with *M. pulcherrima* E1, many DEGs in the plant–pathogen interaction pathway were produced, which induced the defense response of loquat pulp tissues. For example, the upregulation of *CML5*, *CML19*, *CML23* and other genes encoding calcium binding proteins can affect the process of hypersensitivity, cell wall enhancement and stomatal closure in plants. The expression of the transcription factor *WRKY29*, which is related to defense genes, *PTI5*, which encodes a resistance protein, *RPS2*, which is related to plant hypersensitivity, *EDS1*, which is related to programmed cell death, and *RBOHC*, which is involved in coding NADPH oxidase, was upregulated in loquat fruits treated with *M. pulcherrima* E1. These results suggest that antagonistic yeast enhances loquat fruit disease resistance through ROS accumulation, oxidative signal transduction, hypersensitive reaction induction, and reinforcement of plant cell walls and stomatal closure. These findings are consistent with our previous study on *M. pulcherrima* E1, which showed enhanced activities of defense-related enzymes including polyphenol oxidase (PPO), peroxidase (POD), catalase (CAT), ascorbate peroxidase (APX), and phenylalanine ammonia-lyase (PAL) in loquat fruits [10].

#### 4.2.2. Analysis of Secondary Metabolic Pathways

When plants encounter pathogen infection, they activate the biosynthesis of diverse secondary metabolites as part of their defense response. These protective compounds primarily consist of phenylpropanoids, flavonoids, terpenoids, and alkaloids, which accumulate significantly at infection sites to combat microbial invasion [33]. Lignin is an important structural component of the plant cell wall. When plants are infected by pathogens, lignin is synthesized in large quantities and lignifies the cell wall, forming a physical barrier to resist the invasion of pathogens [34]. As shown in Table 4, *M. pulcherrima* E1 treatment caused changes in secondary metabolism-related genes in pulp tissues of loquat. In the biosynthetic pathway of phenylpropanoid, *PER52* and *PER63*, which are involved in the synthesis of the resistant substances P-hydroxyphenyl lignin, guaiacol lignin, 5-hydroxyl guaiacyl lignin and syringyl lignin, the shikimate O-hydroxyl cinnamyl transferase gene (*HST* and *ACT*) and the coffee acyl coenzyme A-O-methyltransferase gene (*ATOMT*) were upregulated to varying degrees. In the flavonoid biosynthesis pathway, *FLS*, *ACT* and *HST* jointly participated in the synthesis of myricetin, quercetin, kaempferol and galangin. *UGT88A1* is involved in the synthesis of phlorizin, and *FLS* and *UGT88A1* were significantly upregulated. The above results showed that *M. pulcherrima* E1 could resist pathogen infection by inducing loquat fruit to synthesize more defensive secondary metabolites, such as lignin and flavonoids.

#### 4.2.3. Analysis of the Related Pathways of Plant Hormone Metabolism

Plant hormones serve crucial functions in both developmental processes and stress responses, regulating not only growth, development, and senescence but also mediating defense signaling pathways during plant immunity [35]. There is a set of complex and intersecting hormone signaling networks in plants, and many hormone-mediated stress-related pathways may have antagonistic and synergistic effects, which can coordinate plant resistance alone or together with other signaling pathways [36]. The hormones related to plant disease resistance include ethylene, jasmonic acid (JA), salicylic acid (SA), brassinosterol (BR), cytokinin (CTK), auxin (IAA) and gibberellin (GA) [37,38]. Salicylic acid can interact with the downstream positive regulator *NPR1* to activate the expression of resistance genes such as *WRKY* transcription factors and disease-initiating proteins, thus enhancing the resistance of plants to pathogens [39]. In the JA signaling pathway, JA binding to the receptor protein COI1 can cause the negative regulator *JAZ* to release the positively regulated transcription factor *MYC2*, thereby initiating downstream expression [40]. In this experiment, the expression level of *the NPR1* gene, a regulatory protein in the SA signal transduction pathway, was downregulated, and the expression of the *TIFY9* and *TIFY10A* genes encoding the negative regulatory factor JAZ enzyme in the synthesis of JA was upregulated. This indicated that the antagonistic yeast inhibited SA- and JA-signaling-related gene expression, suggesting that these pathways may not contribute to loquat fruit resistance induced by the yeast.

The auxin signaling pathway is pivotal in programming growth and development and coordinating immunity in plant [41]. The repressor 1 gene of the *PHYB* activation label (*CYP749A22/BAS1*) is the inactivated gene of brassinolide. Excessive expression of the *BAS1* gene will produce brassinolide C-26 hydroxyl hydroxylate, resulting in a decrease in the content of active brassinolide in plants [42]. Table 5 shows that *M. pulcherrima* E1 treatment caused differential expression of plant hormone metabolism-related genes in pulp tissues of loquat. For example, in the auxin metabolic pathway, the auxin response genes *IAA17* and *AHP1*, which encode histidine phosphate transfer proteins and are involved in cytokinin signal transduction, the cytokinin response regulating gene *ARR3* and the gibberellin receptor-related gene *GID1C* were upregulated to varying degrees. In addition, the expression of *XTH22* and *XTH23*, the key genes in the synthesis and metabolism of BR, was upregulated, while the expression of *CYP749A22*, the inactivated brassinolide gene, was downregulated. This indicated that antagonistic yeast treatment inhibited the expression of genes related to the SA and JA signaling pathways and mainly improved the disease resistance of loquat fruits by activating the expression of IAA, GA, CTK and BR related to signal transduction.

#### 4.2.4. Analysis of Glutathione Metabolic Pathways

Glutathione is a vital antioxidant in plants, primarily functioning to scavenge excess ROS, thereby offering multifaceted protection. It can eliminate excessive reactive oxygen radicals in plants and has detoxification and anti-aging effects, which are closely related to the stress resistance of plants [28,43]. Glutathione S-transferase (GST) is a multifunctional superfamily enzyme that catalyzes the binding interaction between some harmful electron-loving compounds and glutathione. It is the main detoxification system in many plants [44]. It has been reported that GSTs are involved in the accumulation and transport of resistant flavonoid substances in plants [45], and *HaGSTU1* protein can play an important role in improving the resistance of sunflower against *Paracoccus nucleuses* [46]. *APX* also plays a very important role in the plant response to stress. It can protect cells from the toxicity of ROS by removing excessive H_2_O_2_ produced in cells [47]. Studies have shown that inducing the expression of the *APX* gene in poplar trees is closely related to the enhancement of its resistance to the ulcerative bacteria *Botryosphaeria dothidea* [48]. According to Table 6, in the glutathione metabolic pathway, *M. pulcherrima* E1 treatment caused the upregulated expression of *GSTU8* and *GSTU10*, which are involved in coding the key enzyme GSTs catalyzing the glutathione binding reaction. In addition, the expression of *APX3*, a gene involved in encoding ascorbate peroxidase, was also upregulated in the treatment with antagonistic yeast. Our previous study also demonstrated that the activity of ascorbate peroxidase (APX) was significantly enhanced in loquats treated with *Metschnikowia pulcherrima* E1 [10]. This suggests that the antagonistic yeast treatment enhances disease resistance by activating glutathione metabolism on loquat fruits.

## 5. Conclusions

In this study, the pathogen of gray spot disease in loquat was identified as *Pestalotiopsis vismiae* from rotten loquat fruits, and *Metschnikowia pulcherrima* E1 was screened as an efficient antagonistic yeast, which reduced the fruit rot rate to 22.73% through dual mechanisms of iron competition and biofilm formation.

Then, transcriptome sequencing was performed on loquat fruits in the antagonistic yeast *M. pulcherrima* E1 treatment group and the sterile water control group, and 1444 differentially expressed genes were screened out. The results showed that these DEGs were involved in various signaling pathways and metabolic pathways related to fruit resistance, including plant–pathogen interaction, plant hormone metabolism, secondary metabolism, and glutathione metabolism. In addition, the RT–qPCR validation results of DEGs were consistent with the basic trend of sequencing results: the expression levels of *CML19*, *FLS*, *ACT*, *XTH23*, *AFS1*, *CP1*, *HST*, *FHY*, and *GID1C* were upregulated, while the expression levels of *CML42* and *CYP74A22* were downregulated. These findings validate the molecular mechanisms of yeast-induced plant resistance and provide a theoretical basis for biocontrol applications.

## Figures and Tables

**Figure 1 jof-11-00497-f001:**
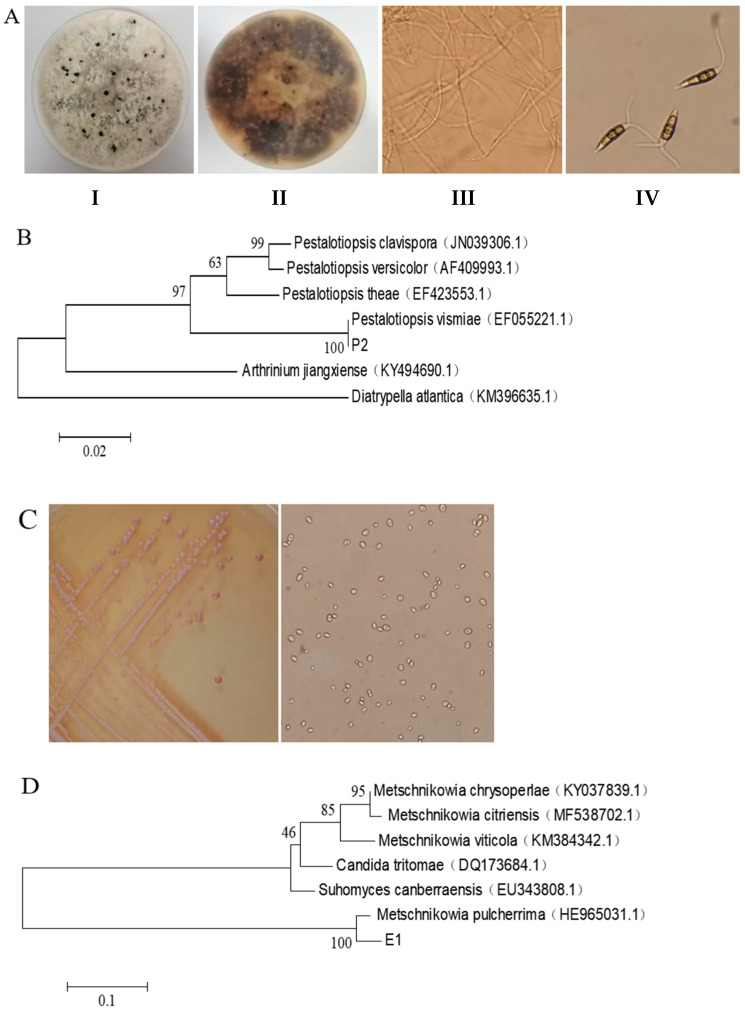
(**A**): Morphological characteristics of strain P2 and strain E1. (**I**) Colony morphology of P2 on the front of a PDA plate; (**II**) colony morphology of P2 on the back of a PDA plate; (**III**) image of mycelia of P2; (**IV**) image of conidia of P2. (**B**): Molecular evolutionary tree of P2 (SAMN49174010). The phylogenetic tree was constructed by the neighbor-joining method using MEGA 5.1 software. The bootstrap values are shown at the branch points. (**C**): Colony (**left**) and cell (**right**) morphology of strain E1. (**D**): Molecular evolutionary tree of *Metschnikowia pulcherrima* E1(SAMN49174009). The phylogenetic tree was constructed by the neighbor-joining method using MEGA 5.1 software. The bootstrap values are shown at the branch points.

**Figure 2 jof-11-00497-f002:**
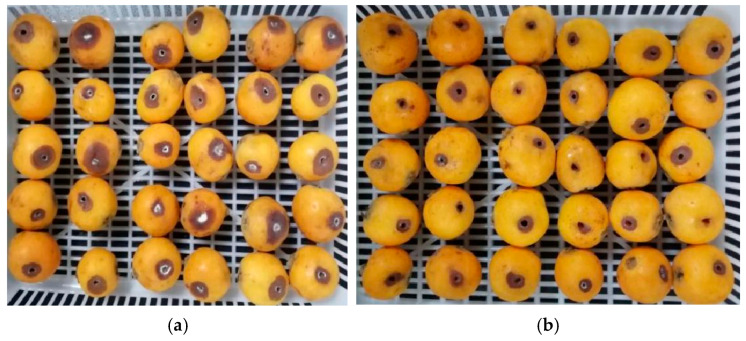
Effect of *M. pulcherrima* E1 on the decay rate of gray spot in loquat fruit. (**a**) Control group: treated with sterile water of one parallel replicate. (**b**) Treatment group: treated with *M. pulcherrima* E1 of one parallel replicate.

**Figure 3 jof-11-00497-f003:**
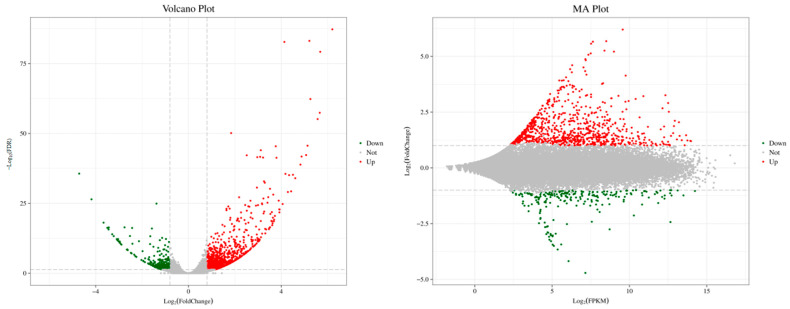
Volcano plot and MA plot of DEGs. The Volcano plot shows the relationship between the significance (*Y*-axis) and fold change (*X*-axis) of differentially expressed genes (DEGs). The MA plot displays the distribution of gene expression levels (*X*-axis) and fold changes (*Y*-axis). Green points represent downregulated DEGs, red points represent upregulated DEGs, and black points represent non-differentially expressed genes.

**Figure 4 jof-11-00497-f004:**
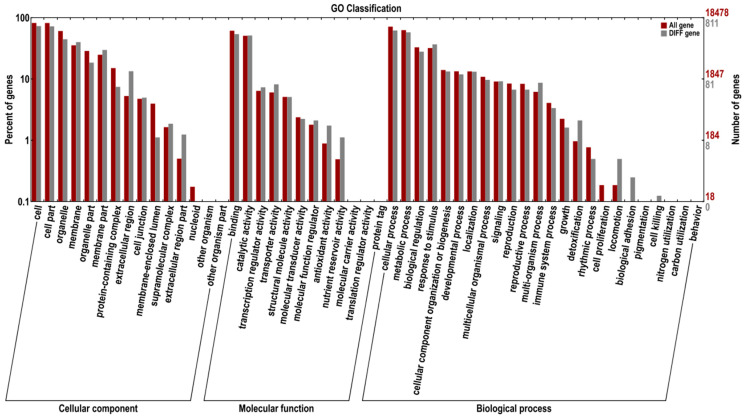
Classification of DEGs based on GO annotation. DIFF gene: differentially expressed gene (DEG), defined as genes with |log_2_(FC)| ≥ 0.8 and FDR < 0.05.

**Figure 5 jof-11-00497-f005:**
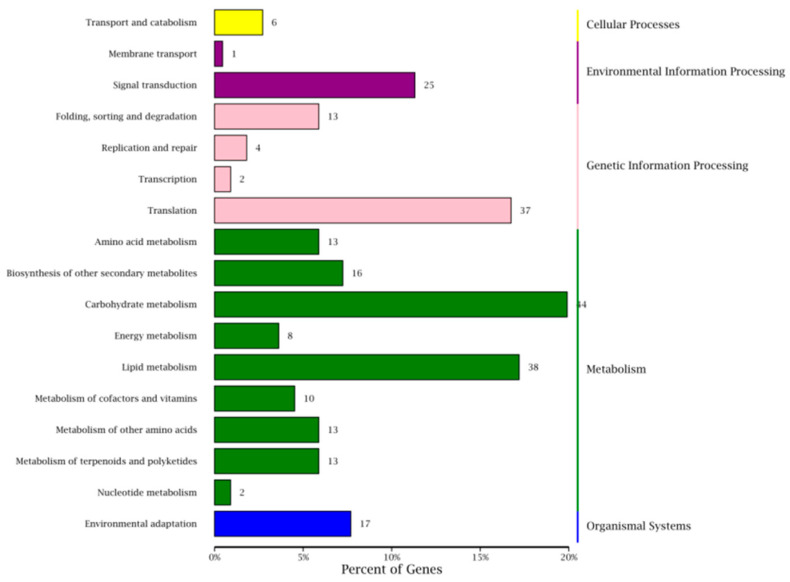
KEGG pathway classification of DEGs.

**Figure 6 jof-11-00497-f006:**
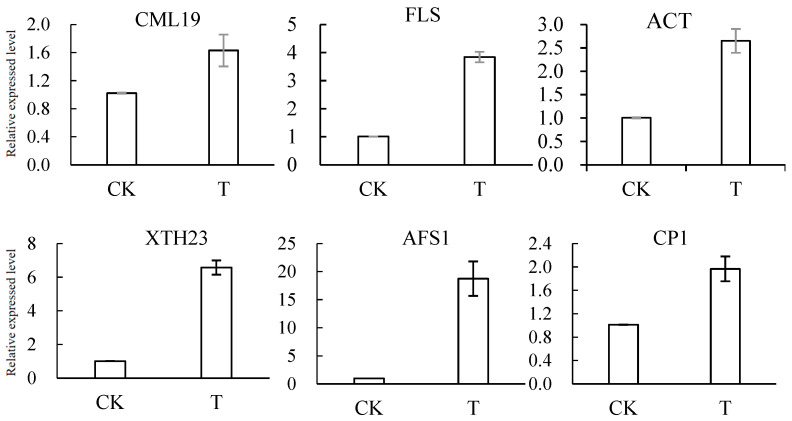
Results of verification of some DEGs by RT–qPCR.

**Table 1 jof-11-00497-t001:** Effect of *M. pulcherrima* E1 on the incidence and diameter of gray spot in loquat.

**Experiment**	**Incident Rate (%)**	**Spot Diameter (cm)**
treatment group	22.73	0.95 ± 0.06 ^b^
control group	100	1.59 ± 0.16 ^a^

Different letters in the same column indicate significant differences (*p* < 0.05). The standard error of the treatment group is 0.01, and that of the control group is 0.03 (The standard error is calculated by dividing the standard deviation by the square root of the sample size, *n* = 30).

**Table 2 jof-11-00497-t002:** Sequencing data tables of loquat tissues.

Samples	JSHY-ID	ReadSum	BaseSum	GC (%)	Q30 (%)
CK1	T01	24562436	7368730800	46.56	92.22
CK2	T02	25701745	7710523500	46.03	92.84
CK3	T03	23718990	7115697000	46.5	92.95
T1	T04	28165336	8449600800	46.44	94.37
T2	T05	24068412	7220523600	46.48	94.24
T3	T06	23280409	6984122700	46.26	92.62

**Table 3 jof-11-00497-t003:** Effects of *M. pulcherrima* E1 on loquat genes related to plant–pathogen interactions.

Gene ID	Gene Name	log_2_FC	Definition
TRINITY_DN14842_c0_g1	*RBOHC*	0.96	respiratory burst oxidase
TRINITY_DN12258_c0_g1	*CML5*	1.40	calcium-binding protein
TRINITY_DN14083_c0_g1	*HSP82*	−1.16	molecular chaperone HtpG
TRINITY_DN15797_c0_g1	*CML23*	1.56	calcium-binding protein
TRINITY_DN15820_c0_g1	*CML27*	1.57	calcium-binding protein
TRINITY_DN15820_c0_g3	*CML27*	1.17	calcium-binding protein
TRINITY_DN16219_c0_g1	*CML19*	2.00	calcium-binding protein
TRINITY_DN16219_c0_g2	*CML19*	0.97	calcium-binding protein
TRINITY_DN17371_c0_g3	*EDS1*	1.53	enhanced disease susceptibility 1 protein
TRINITY_DN17724_c5_g2	*CP1*	1.20	Calcium-binding protein
TRINITY_DN17724_c5_g8	*CP1*	1.74	Calcium-binding protein
TRINITY_DN18123_c0_g1	*CML42*	−1.64	calcium-binding protein
TRINITY_DN18265_c0_g4	*CML36*	1.04	calcium-binding protein
TRINITY_DN24035_c0_g1	*RPS2*	1.06	disease resistance protein
TRINITY_DN20679_c0_g1	*WRKY29*	0.86	WRKY transcription factor 29
TRINITY_DN23615_c1_g2	*PTI5*	0.93	pathogenesis-related genes transcriptional activator

**Table 4 jof-11-00497-t004:** Effects of *M. pulcherrima* E1 on loquat genes related to secondary metabolism. All log_2_FC values represent statistically significant differentially expressed genes (|log_2_FC| ≥ 0.8, FDR < 0.05), as defined in Figure 3.

Gene ID	Gene Name	log_2_FC	Definition
Phenylpropanoid biosynthesis			
TRINITY_DN15146_c0_g1	*PER52*	1.82	peroxidase
TRINITY_DN15166_c0_g1	-	0.97	caffeoyl-CoA O-methyltransferase
TRINITY_DN15653_c0_g1	*PER52*	1.23	peroxidase
TRINITY_DN16201_c0_g1	*MEE23*	−1.06	cinnamyl-alcohol dehydrogenase
TRINITY_DN17214_c1_g1	*ACT*	1.55	shikimate O-hydroxycinnamoyltransferase
TRINITY_DN18626_c3_g2	*PER63*	0.92	peroxidase
TRINITY_DN22855_c1_g4	*HST*	1.06	shikimate O-hydroxycinnamoyltransferase
TRINITY_DN23721_c0_g2	*F6&apos*; *H1*	−1.52	feruloyl-CoA 6-hydroxylase
Flavonoid biosynthesis			
TRINITY_DN15166_c0_g1	-	0.97	caffeoyl-CoA O-methyltransferase
TRINITY_DN17214_c1_g1	*ACT*	1.55	shikimate O-hydroxycinnamoyltransferase
TRINITY_DN21813_c0_g1	*PKS5*	0.88	chalcone synthase
TRINITY_DN21935_c3_g2	*SALAT*	−0.88	shikimate O-hydroxycinnamoyltransferase
TRINITY_DN22199_c0_g1	*UGT88A1*	1.10	phlorizin synthase
TRINITY_DN22855_c1_g4	*HST*	1.06	shikimate O-hydroxycinnamoyltransferase
TRINITY_DN9789_c0_g1	*FLS*	1.26	flavonol synthase
Diterpenoid biosynthesis			
TRINITY_DN15372_c0_g1	*GA2OX8*	1.36	gibberellin 2beta-dioxygenase
TRINITY_DN15922_c0_g2	*GA2OX1*	1.80	gibberellin 2beta-dioxygenase
TRINITY_DN20240_c0_g1	*GA2OX2*	0.97	gibberellin 2beta-dioxygenase
TRINITY_DN6982_c0_g1	*GES*	1.05	geranyllinalool synthase
Sesquiterpenoid and triterpenoid biosynthesis			
TRINITY_DN12696_c0_g1	*AFS1*	2.38	alpha-farnesene synthase
TRINITY_DN22548_c2_g1	-	−1.01	squalene monooxygenase
Riboflavin metabolism			
TRINITY_DN10940_c0_g1	*PAP3*	2.91	tartrate-resistant acid phosphatase type 5
TRINITY_DN12491_c0_g1	*FHY*	2.59	riboflavin kinase/FMN hydrolase
TRINITY_DN14884_c0_g1	*PAP17*	3.41	tartrate-resistant acid phosphatase type 5
Stilbenoid, diarylheptanoid and gingerol biosynthesis			
TRINITY_DN15166_c0_g1	-	0.97	caffeoyl-CoA O-methyltransferase
TRINITY_DN17214_c1_g1	*ACT*	1.55	shikimate O-hydroxycinnamoyltransferase
TRINITY_DN21935_c3_g2	*SALAT*	−0.88	shikimate O-hydroxycinnamoyltransferase
TRINITY_DN22855_c1_g4	*HST*	1.06	shikimate O-hydroxycinnamoyltransferase

**Table 5 jof-11-00497-t005:** Effects of *M. pulcherrima* E1 on loquat genes related to phytohormone metabolism.

Gene ID	Gene Name	log_2_FC	Definition
Auxin metabolism			
TRINITY_DN26772_c0_g1	*IAA17*	1.14	auxin-responsive protein IAA
TRINITY_DN14799_c0_g1	*SAUR32*	−0.84	SAUR family protein
Cytokinine metabolism			
TRINITY_DN21313_c0_g1	*AHP1*	0.81	histidine-containing phosphotransfer protein
TRINITY_DN17952_c2_g1	*ARR3*	0.94	two-component response regulator ARR-A
Gibberellin metabolism			
TRINITY_DN13496_c0_g1	*GID1C*	1.00	gibberellin receptor GID1
Brassinosteroid biosynthesis and metabolism			
TRINITY_DN18076_c3_g4	*CYP749A22*	−1.00	PHYB activation tagged suppressor 1
TRINITY_DN20436_c1_g2	*CYP749A22*	−1.11	PHYB activation tagged suppressor 1
TRINITY_DN17619_c0_g1	*XTH23*	2.95	xyloglucan:xyloglucosyl transferase TCH4
TRINITY_DN19437_c1_g1	*XTH22*	1.51	xyloglucan:xyloglucosyl transferase TCH4
TRINITY_DN21110_c1_g4	*XTH23*	3.79	xyloglucan:xyloglucosyl transferase TCH4
TRINITY_DN21863_c1_g1	*XTH22*	2.63	xyloglucan:xyloglucosyl transferase TCH4
Jasmonic acid			
TRINITY_DN16088_c0_g2	*TIFY10A*	1.86	jasmonate ZIM domain-containing protein
TRINITY_DN17853_c2_g4	*TIFY9*	1.58	jasmonate ZIM domain-containing protein
Salicylic acid			
TRINITY_DN14251_c0_g1	*NPR1*	−1.29	regulatory protein NPR1

**Table 6 jof-11-00497-t006:** Effects of *M. pulcherrima* E1 on loquat genes related to glutathione metabolism.

Gene ID	Gene Name	log_2_FC	Definition
TRINITY_DN17157_c0_g3	-	0.94	glutathione S-transferase
TRINITY_DN17817_c0_g1	-	0.82	glutathione S-transferase
TRINITY_DN14083_c0_g1	-	2.18	glutathione S-transferase
TRINITY_DN21103_c2_g4	-	0.97	glutathione S-transferase
TRINITY_DN21103_c2_g5	-	3.23	glutathione S-transferase
TRINITY_DN21174_c0_g1	-	0.82	glutathione S-transferase
TRINITY_DN22846_c1_g1	*CDKD-1*	0.94	glutathione S-transferase
TRINITY_DN24039_c0_g1	*GSTU10*	3.60	glutathione S-transferase
TRINITY_DN11198_c0_g1	*APX3*	1.58	L-ascorbate peroxidase

## Data Availability

The data shown in this study are contained within the article.

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
