# Peer review of "The Study of *Metschnikowia pulcherrima* E1 in the Induction of Improved Gray Spot Disease Resistance in Loquat Fruit"

_jof, 2025, doi:10.3390/jof11070497_

Round 1

Reviewer 1 Report

  • English must be revised in the text.
  • In the abstract, the main results of the work should be provided and therefore it needs to be revised.
  • The aim of the work should be expressed more clearly in the introduction.
  • I suggest to the authors to include an experimental plan in graphic form explaining the work organisation in a simple manner.
  • Insert a section relating of applied statistical analysis
  • The acronym DEG should also be expressed in full in the introduction the first time it is mentioned.

Reviewer 2 Report

The paper entitled 'The study of Metschnikowia pulcherrima E1 in the induction of improved gray spot disease resistance in loquat fruit' is devoted to the study of mechanism of inhibition of Pestalotiopsis vismiae by the yeast Metschnikowia pulcherrima. Biocontrol organisms have been considered as a promising alternative to fungicides. Therefore, the paper is timely and scientifically significant. At the same time, the paper's text should be seriously improved, first of all in 'molecular genetic' parts. 

  1. I couldn't find any Supplementary files, despite the authors mentioned Table S1 (Line 138).
  2. I think the authors should not describe what is transcriptomics such in detail in the Introduction. Instead of this, they could more comprehensively discuss the use of biocontrol agents against Pestalotiopsis spp. (not only P. vismiae). What orgaisms were used to control these pathogens, their efficiencies, mechanisms of action. 
  3. 2.2.: the authors mentioned that 'pathogen was identified by morphology and molecular biology'. What does it mean? If the authors sequenced a specific fragment for identification, then DNA isolation protocol, target gene, primer sequences, amplification conditions and sequencing protocol should be provided.
  4. 2.3. The same: DNA isolation protocol, PCR conditions and primer sequences are missing. 
  5. Phylogenetic analysis techniques should also be described in 'Materials in Methods' 
  6. 2.5.2, ; line 122 - what RNA kit was used? What was the incubation time when RNA was extracted?
  7. 2.7. Primer structures as well as algorithms of relative expression calculation are missing.
  8. 3.1., Fig. 1. The sequences should be deposited in GenBank and accession numbers should be provided.
  9.  3.7. The authors should describe the results of qPCR analysis of DEGs more comprehensively. Particularly, relative expression for each gene (fold changes) can be mentioned, and the Fig. 6 should be described more in detail. 
  10. I believe that in Discussion the authors should also discuss the use of M. pucherrima as biocontrol agent (for instance, mention Puyo et al., 2023; El Dana et al., 2025; Bustamante et al., 2025).

Line 28: 'Resistance' should be among the keywords. 

Line 42: not 'resistance to pathogens' but 'pathogen resistance to fungicides' 

2.7. What gene was used as a reference?

Line 161: MegA 5.1.

Line 403: 'Glutathione' repeats twice

Reviewer 3 Report

Dear Authors, dear Editor,

This study provides valuable insights by expanding upon previous findings, particularly those cited in reference [4]. Through inoculation on detached fruits and mechanical wounding, you performed transcriptome sequencing and validated the expression of 11 genes. While the work is compelling, I offer the following suggestions to further strengthen the manuscript:

  1. Pathogenicity of DEGs:

Given the possible stress responses induced by fruit detachment and mechanical wounding, I recommend discussing how these factors may affect the observed differentially expressed genes (DEGs) and their association with pathogenicity. To address this, you could include transcriptomic data from non-wounded fruits as a control or reinforce your interpretations with strong literature-based arguments.

  1. Connection to Reference [4]:

Please elaborate on the relationship between your transcriptome analysis and the results reported in reference [4], particularly regarding volatile organic compounds (3-methyl-1-butanol, phenylethyl alcohol, 1-hexanol) and defense-related enzymes (polyphenol oxidase, peroxidase, catalase, ascorbate peroxidase, phenylalanine ammonia-lyase) produced by M. pulcherrima E1.

These two aspects directly relate to the scope suggested by the current title. As it stands, the title would be more justified if you had demonstrated reduced disease symptoms in greenhouse or field trials under unwounded conditions. However, given the current evidence—differential expression in at least 11 validated genes—I suggest revising the title to reflect the main findings (i.e., transcriptome analysis).

  1. Abstract: Please highlight the most relevant and significant findings in the Abstract, particularly those supported by transcriptome validation (e.g., Figure 6).
  1. Introduction: Clarify the knowledge gap that motivated this study, especially in connection with your prior work (reference [4]). Explicitly state how this study advances the understanding established in that publication.
  1. Conclusions: Strengthen this section by concisely summarizing the most critical findings, with emphasis on the alignment between the transcriptomic data and the gene expression validation shown in Figure 6.
  1. Pathogen Sequencing: Please specify which genes were sequenced for pathogen identification, as this is currently omitted. Additionally, provide a rationale for why ITS sequencing alone is sufficient to resolve the identity at the species level.
  2. Figures & Tables (self-explanatory):
    1. Figure 3: Clearly label both the Volcano and MA plots (include axis labels, meaning of points, and explain the left/right panels).
    2. Figure 4: Clarify the meaning of “DIFF gene.”
    3. Figure 6: Add appropriate units to the Y-axis.
    4. Table 4: Indicate whether the reported log2FC values are statistically significant (as shown in Figure 3). If all values are significant, state this explicitly.
  3. Style Edits: Lines 337–338: The phrase “the rapid defense response” may be misleading, as a response observed at 3 dpi (days post-inoculation) is not typically considered rapid.
  4. Subsection 3.6.1: Remove the duplicated word "pathogen" from the heading.

Reviewer 4 Report

The document is very interesting; however, I believe it needs corrections to be considered for publication.
Its objective is unclear.
It does not describe key points, such as the number of fruits used for the study and how many per group. Furthermore, it is not known how many groups there were because they were never described.
It does not describe the statistical analysis used for its work.
It presents its figures as AA, not A1, so it is suggested that it be modified.
It provides a detailed description of its results in the initial section of its discussion; however, I believe this should be presented in its results.
It summarizes the findings of its work in the conclusion section, but does not provide clear conclusions.

The document is very interesting; however, I believe it needs corrections to be considered for publication.
Its objective is unclear.
It does not describe key points, such as the number of fruits used for the study and how many per group. Furthermore, it is not known how many groups there were because they were never described.
It does not describe the statistical analysis used for its work.
It presents its figures as AA, not A1, so it is suggested that it be modified.
It provides a detailed description of its results in the initial section of its discussion; however, I believe this should be presented in its results.
It summarizes the findings of its work in the conclusion section, but does not provide clear conclusions.

Round 2

Reviewer 1 Report

Authors responded satisfactorily to my suggestions, so the work is definitely improved.

I have no other suggestions.

Reviewer 2 Report

In the revised version of the paper, the authors took into account the majority of comments made. I believe that the current version is close to be ready for acceptance. At the same time, some points are still to be improved (see 'Detailed comments').

Lines 35-37: References?

Line 84: 'determine'

Line 136: the sequence of 5.8S rDNA-ITS region of E1 was identified by Sanger sequencing' (if it was so), not 'molecular biology' 

Section 2.2. and 2.3.: a. Accession numbers for ITS sequences? b. What equipment was used for PCR amplification?

Section 2.7. Did the authors use a DNA-binding dye such as SYBR green or oligonucteotide probes for fluorescene detection?

Line 232: 'MEGA'

Reviewer 4 Report

Thank you for considering the comments to improve your work.
However, I believe it still needs some corrections before publication.
1. Add the standard error of your treatments to Table 1.
2. Use the same sections of your work in the discussion as you did in the results, which will help better understand your results.

Thank you for considering the comments to improve your work.
However, I believe it still needs some corrections before publication.
1. Add the standard error of your treatments to Table 1.
2. Use the same sections of your work in the discussion as you did in the results, which will help better understand your results.
